

# A convenient diagnostic tool for discriminating adult-onset glutamic acid decarboxylase antibody-positive autoimmune diabetes from type 2 diabetes: a retrospective study

Hon-Ke Sia[1,2], Shih-Te Tu[1], Pei-Yung Liao[1], Kuan-Han Lin[2], Chew-Teng Kor[3] and Ling-Ling Yeh[4]

[1] Division of Endocrinology and Metabolism, Department of Internal Medicine, Changhua Christian Hospital, Changhua City, Taiwan
[2] Department of Healthcare Administration, Asia University, Taichung City, Taiwan
[3] Internal Medicine Research Center, Changhua Christian Hospital, Changhua City, Taiwan
[4] Social Enterprise and Innovation M.A. Program, Dharma Drum Institute of Liberal Arts, New Taipei City, Taiwan

Corresponding author
Ling-Ling Yeh, YehLL@dila.edu.tw

## ABSTRACT

**Background.** The glutamic acid decarboxylase antibody (GADA) test, commonly used to diagnose autoimmune diabetes, is not cost-effective in areas of low prevalence. The aim of this study was to develop a convenient tool to discriminate adult-onset GADA-positive autoimmune diabetes from type 2 diabetes (T2DM) in patients with newly diagnosed diabetes.

**Methods.** This retrospective cross-sectional study, conducted at Changhua Christian Hospital in Taiwan, collected electronic medical record data from January 2009 to December 2018. Patients were divided into a case group (GADA+, $n = 152$) and a reference group (T2DM, $n = 358$). Variables that differed significantly between the groups were subjected to receiver operator characteristic analysis to establish cutoff values. Discriminant function analysis was then employed to discriminate the two groups.

**Results.** At the onset of diabetes, the GADA+ group was younger, with lower body mass index (BMI), higher hemoglobin A1c (HbA1c), higher high-density lipoprotein cholesterol (HDL-C), and lower total cholesterol and triglycerides (TG). Five major factors were identified to form the linear discriminant functions: BMI, age at onset, TG, HDL-C, and HbA1c. BMI $< 23$ kg/m$^2$ was the most important factor, followed by TG $< 98$ mg/dL, HDL-C $\geq 46$ mg/dL, age at onset $< 30$ years, and HbA1c $\geq 8.6\%$. The overall accuracy of the linear discriminant functions was 87.1%, with 84.2% sensitivity and 88.3% specificity.

**Conclusions.** Routine tests in diabetes care were used to establish a convenient, low-cost tool that may assist in the early identification of adult-onset GAD+ autoimmune diabetes in clinical practice.

## INTRODUCTION

In children and adolescents, type 1 diabetes (T1DM), also known as autoimmune diabetes, usually presents with diabetic ketoacidosis (DKA) as the initial presentation and is relatively straightforward to diagnose. However, the clinical features of adult-onset autoimmune diabetes can be less typical. The period of insulin requirement after onset may allow differentiation between patients with latent autoimmune diabetes in adults (LADA) and those with classic adult-onset T1DM, who become insulin-dependent within 3 months of diagnosis (*Buzzetti, Zampetti & Maddaloni, 2017*). LADA often involves slow, progressive islet destruction. In the absence of autoantibody testing, it is not uncommon for it to be diagnosed and treated as type 2 diabetes (T2DM) (*Buzzetti, Zampetti & Maddaloni, 2017*; *Carlsson, 2019*). The lack of optimal treatment can result in deterioration of the autoimmune process, the acceleration of beta cell loss, faster progression to insulin dependence, and an increased risk of complications (*Carlsson, 2019*; *Pieralice & Pozzilli, 2018*).

Testing of islet autoantibodies is essential to diagnose autoimmune diabetes. Islet cell cytoplasmic autoantibodies (ICA), insulin autoantibodies (IAA), glutamic acid decarboxylase antibodies (GADA), tyrosine phosphatase IA-2 autoantibodies (IA-2A), and zinc transporter 8 autoantibodies (ZnT8A) are the major autoantibodies of clinical and research interest (*Buzzetti, Zampetti & Maddaloni, 2017*; *Winter & Schatz, 2011*). The reported incidence of T1DM-related autoantibodies in adult-onset diabetes is approximately 3%–12%, with variations between countries and ethnicities (*Buzzetti, Zampetti & Maddaloni, 2017*; *Carlsson, 2019*; *Pieralice & Pozzilli, 2018*). GADA are persistent in patients with long-standing diabetes and is not influenced by the age at disease onset. Therefore, the presence of GADA represents the most sensitive marker in adult-onset autoimmune diabetes (*Buzzetti, Zampetti & Maddaloni, 2017*). However, the GADA test has not been widely used in primary care because of its cost and limited availability. In areas with a low prevalence of autoimmune diabetes, GADA measurement is not cost-effective as a routine test and can only be used in highly selective cases. It would therefore be desirable to establish a practical tool that could be used to determine whether to test GADA for suspected patients.

Fourlanos et al. proposed a LADA clinical risk score as a clinical screening tool to identify patients with LADA (*Fourlanos et al., 2006*). The parameters included age at onset <50 years, acute symptoms, body mass index (BMI) <25 kg/m$^2$, and a personal or family history of autoimmune disease. According to a study by Shields, the two most important indicators for identifying T1DM and T2DM are age at diagnosis <35 years and time to insulin <6 months (*Shields et al., 2015*). These studies showed that age at diagnosis and BMI were important discriminators. Conversely, *Thunander et al. (2012)* demonstrated that C-peptide level at diagnosis was better than age or BMI in discriminating between autoimmune and non-autoimmune diabetes. However, C-peptide measurement is also limited by its cost and availability.

The metabolic and clinical phenotypes of adult-onset autoimmune diabetes are heterogeneous, ranging from patients who are lean and insulin sensitive to those who

are obese and insulin resistance (*Buzzetti, Zampetti & Maddaloni, 2017*). There is evidence that the various disease processes observed in subtypes of autoimmune diabetes can be detected by different autoantibodies. Diabetic patients positive for the IA-2$_{(256-760)}$ autoantibody showed slow beta cell deterioration and clinical and metabolic phenotype, similar to those with classic type 2 diabetes and obesity (*Buzzetti, Zampetti & Maddaloni, 2017*; *Pieralice & Pozzilli, 2018*). Conversely, the A Diabetes Outcome Progression Trial (ADOPT) reported that GADA+ diabetic patients had higher levels of serum high-density lipoprotein cholesterol (HDL-C) and lower levels of serum triglycerides (TG) than those diagnosed T2DM, as well as a lower prevalence of metabolic syndrome (*ADOPT Study Group, 2004*). A Korean population study found that GADA levels were inversely associated with age at onset, C-peptide levels, BMI, and total cholesterol and TG concentrations (*Roh et al., 2013*). These studies demonstrated that the lipid profile can help to distinguish autoimmune diabetes with GADA + from T2DM.

Several related studies have used multiple islet autoantibodies to detect autoimmune diabetes rather than focusing on GADA (*Buzzetti, Zampetti & Maddaloni, 2017*; *Carlsson, 2019*; *Pieralice & Pozzilli, 2018*). Subtype studies pointing to GADA+ diabetes are therefore valuable in providing additive information for managing adult-onset autoimmune diabetes. In addition, there is still some debate on the definition of LADA as well as on the classification of type 1 diabetes mellitus (*Carlsson, 2019*). Using GADA+ diabetes in the present study would help in defining the research subjects more distinctly and precisely.

The aim of the present study was to develop a convenient, cost-effective tool that can be used to discriminate adult-onset GADA+ autoimmune diabetes from T2DM. Such a tool could help physicians decide whether to perform a GADA test. To avoid additional costs, we based the tool on parameters obtained from diabetes routine tests, such as lipid profile and hemoglobin A1c (HbA1c).

## MATERIALS & METHODS

### Subjects

This retrospective cross-sectional study was conducted at Changhua Christian Hospital (CCH), a medical center in Taiwan. A total of 636 patients with diabetes who received the GADA test between January 1, 2009 and December 31, 2018 were screened for eligibility based on data from the hospital's electronic medical record system. Among these, 484 patients were excluded for the following reasons: with GADA <1.0 U/mL, incomplete lipid and A1c data, diabetes not diagnosed within the previous six months, age at onset <20 years, and excessive alcohol consumption. Finally, 152 patients were enrolled into the case group (GADA+ group). Excessive alcohol consumption was defined as consuming 7 drinks, amounting to approximately 70 grams of alcohol, or more per week for both men and women.

The reference group (T2DM group) included 358 type 2 diabetic patients, newly diagnosed between January 1, 2010 and December 31, 2013, with age at onset ≥ 20 years. GADA was not a routine test in the reference group. However, GADA testing was selectively performed for patients suspected of having autoimmune diabetes in follow-up visits. Two

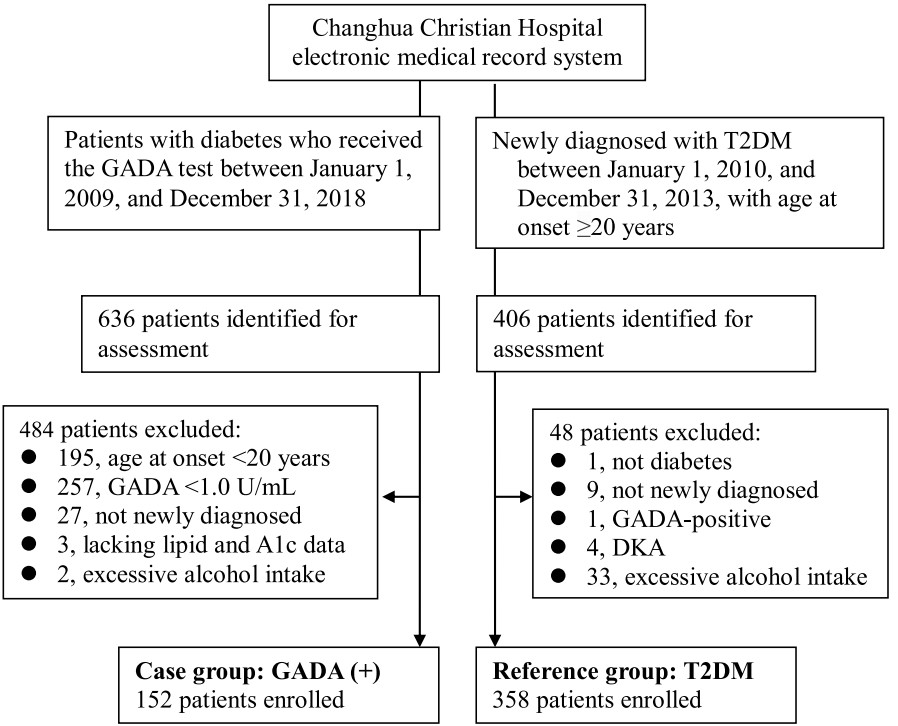

**Figure 1** **Enrollment of participants in the study.** Abbreviations: GADA, glutamic acid decarboxylase antibody; T2DM, type 2 diabetes mellitus; DKA, diabetic ketoacidosis.

diabetologists reviewed the medical records from January 1, 2010 to December 31, 2018, for each patient of the reference group to confirm the diagnosis and to reduce the possibility of misclassification to the maximum extent. Thus, 15 were excluded from the 406 patients identified for assessment (Fig. 1).

## Data collection

The data for both groups were obtained from the hospital's electronic medical record system. Basic data (age at the onset of diabetes, and gender), personal habits (smoking and drinking), height, and body weight were those collected at the initial visit for the diagnosis of diabetes. The laboratory data used in the analysis, including HbA1c, creatinine (Cr), and glutamic pyruvic transaminase (GPT) levels, were those measured closest to the first visit. The lipid profile selected, including total cholesterol, HDL-C and TG levels, was that measured 2–12 weeks after the first visit, documenting the use of any lipid-lowering agents (fibrates or statins) before the lipid profile test.

## Measurements

The diagnosis of diabetes mellitus was made according to the American Diabetes Association criteria of 2019 (*American Diabetes Association, 2019*). Two diabetologists with more than five years' clinical experience reviewed the medical records to validate the diagnosis. In cases of disagreement, the patient was excluded. The total amount of alcohol consumption

in grams per week was calculated from weekly alcohol consumption (in millimeter) and the alcoholic concentration of each kind of beverage.

GADA was measured with an anti-GAD radioimmunoassay kit (RSR Ltd., Cardiff, UK). According to the manufacturer, a serum concentration of GADA ≥1 u/ml was considered to be positive for GADA. The specificity of the GADA test for diagnosing adult autoimmune diabetes can be as high as 98%, with a sensitivity of about 70%–80% (*Wenzlau et al., 2007*).

Serum biochemistry, including glucose, Cr, and GPT levels and the lipid profile, was performed using a UniCel DxC 800 SynchronTM Clinical System (Beckman Coulter, Brea, CA, USA). Serum HbA1c was measured by ion-exchange high-performance liquid chromatography using a VARIANTTM II Turbo system.

BMI was calculated as the body weight in kilograms divided by height in meters squared. A BMI of 23.0–24.9 kg/m$^2$ was classified as overweight and ≥25 kg/m$^2$ as indicating obesity, according to the Regional Office for the Western Pacific standard (*World Health Organization, Regional Office for the Western Pacific (WPRO), International Association for the Study of Obesity, International Obesity Task Force, 2000*).

## Statistical analyzes

To establish a convenient diagnostic tool for discriminating the case group (GADA+) from the reference group (T2DM), the major discriminating factors were first identified and then used to form discriminant functions by discriminant function analysis.

Step 1: Differences between the two groups for selected variables were evaluated by Student's *t*-test for continuous variables and chi-square test for categorical variables. The continuous variables with statistical significance were tested for normal distribution using the Kolmogorov–Smirnov test, and data with *P*-values of <0.05 were regarded as non-normally distributed. The non-normally distributed variables were transformed to ordinary variables for further analysis. A cutoff value for each of these variables was determined from the point on the receiver operating characteristic (ROC) curve with the minimum distance to the upper left corner, calculated as the square root of $[(1 - \text{sensitivity})^2 + (1 - \text{specificity})^2]$.

Step 2: Discriminant function analysis using the enter method was then applied to establish the major discriminating variables according to the discriminant loadings (with absolute values >0.3) in the structure matrix. These major discriminating variables were then analyzed, again with the enter method, to determine the standardized canonical discriminant function.

All the tests were two-tailed with a significance level of 0.05. SPSS version 25 software (IBM Corp., Armonk, NY, USA) was used for the analysis.

## Ethics statement

The study was approved by the Institutional Review Board of Changhua Christian Hospital (CCH IRB No: 190702). Informed consent was waived.

## RESULTS

This study included a total of 510 Chinese patients who were ethnically homogeneous. In the GADA+ group, 84 patients (55.3%) had higher GADA titers (≥32 U/mL) than the

**Table 1  Clinical features of the patients at the diagnosis of diabetes mellitus.**

| | GADA+ (n = 152) Mean ± SD | T2DM (n = 358) Mean ± SD | p |
|---|---|---|---|
| Gender: Men, n (%) | 74 (48.7) | 202 (56.4) | 0.109[a] |
| Age at onset (years) | 37.6 ± 12.6 | 50.4 ± 11.7 | <0.001 |
| 20–29, n (%) | 48 (31.6) | 18 (5.0) | |
| 30–49, n (%) | 75 (49.3) | 133 (37.2) | |
| ≥50, n (%) | 29 (19.1) | 207 (57.8) | |
| BMI (kg/m$^2$) | 21.7 ± 3.8 | 26.8 ± 4.5 | <0.001 |
| <23, n (%) | 114 (75.0) | 62 (17.3) | |
| 23–24.9, n (%) | 16 (10.5) | 70 (19.6) | |
| ≥25, n (%) | 22 (14.5) | 226 (63.1) | |
| Lipid profile | | | |
| Total cholesterol (mg/dL) | 184.0 ± 46.9 | 194.4 ± 41.9 | 0.014 |
| TG (mg/dL) | 82.9 ± 65.2 | 162.2 ± 119.9 | <0.001 |
| HDL-C (mg/dL) | 58.0 ± 18.7 | 42.7 ± 10.5 | <0.001 |
| HbA1c (%) | 10.5 ± 3.2 | 8.2 ± 2.3 | <0.001 |
| GPT (U/L) | 25.1 ± 18.7 | 35.2 ± 29.2 | <0.001 |
| Creatinine (mg/dL) | 0.8 ± 0.2 | 0.8 ± 0.2 | 0.734 |
| Smoking, n (%) | 33 (21.7) | 64 (17.9) | 0.313[a] |
| Use of statins, n (%) | 5 (3.3) | 6 (1.7) | 0.317[b] |
| Use of fibrates, n (%) | 1 (0.7) | 5 (1.4) | 0.675[b] |

**Notes.**
[a] Chi-square test.
[b] Fisher's exact test.
Other variables were examined by $t$-test.
Abbreviations: GADA+, Glutamic acid decarboxylase antibody-positive group; T2DM, type 2 diabetes mellitus group; BMI, body mass index; HDL-C, high-density lipoprotein cholesterol; TG, triglycerides; HbA1c, hemoglobin A1c; Cr, creatinine; GPT, glutamic pyruvic transaminase.

others. Table 1 summarizes the clinical features of the patients at the time of diagnosis of diabetes. The GADA+ group was significantly younger, with lower BMI, higher HbA1c and HDL-C levels, and lower total cholesterol, TG and GPT levels at onset of DM. There were no differences between the two groups in the gender ratio, Cr levels, prevalence of smoking, or the use of lipid-lowering agents (fibrates or statins).

ROC curves were used to determine the cutoff values of these variables for discriminating the patients in the GADA+ group, as follows: HbA1c ≥8.6%, HDL-C ≥46 mg/dL, total cholesterol <183 mg/dL, TG <98 mg/dL, and GPT ≥22 U/L (Table 2). ROC curve figures are shown in Fig. S1.

The discriminant function analysis identified five major discriminating variables from the structure matrix: BMI, age at onset, TG, HDL-C, and HbA1c. Conversely, use of statins, use of fibrates, GPT and total cholesterol were excluded because their discriminant loadings were <0.3. The remaining five major variables (BMI, age at onset, TG, HDL-C, and HbA1c) were analyzed, again by using the enter method, to form linear discriminant functions (Table 3). In order of the absolute magnitude of the correlation within the function, BMI <23 kg/m$^2$ was the most important factor, followed by TG <98 mg/dL, HDL-C ≥46 mg/dL,

**Table 2  Receiver operating characteristic analysis of variables for differentiating the GADA+ from the T2DM patients.**

| Variables | Cut-off value | Sensitivity | Specificity | Minimum distance[a] | AUC | 95% CI |
|---|---|---|---|---|---|---|
| HbA1c (%) | 8.6 | 67.1% | 65.6% | 0.226 | 0.72 | 0.67–0.77 |
| Total cholesterol (mg/dL) | 183 | 57.2% | 58.9% | 0.351 | 0.59 | 0.53–0.65 |
| HDL-C (mg/dL) | 46 | 74.3% | 70.1% | 0.155 | 0.78 | 0.73–0.83 |
| TG (mg/dL) | 98 | 77.6% | 70.7% | 0.136 | 0.80 | 0.76–0.85 |
| GPT (U/L) | 22 | 55.3% | 65.6% | 0.318 | 0.66 | 0.60–0.71 |

Notes.

[a]The minimum distance between a point on the receiver operating characteristic curve and the upper left corner, calculated as the square root of $[(1 - sensitivity)^2 + (1 - specificity)^2]$. The point with the minimum distance was used to define the cutoff value.

Abbreviations: GADA, Glutamic acid decarboxylase antibody; T2DM, type 2 diabetes mellitus; AUC, area under the receiver operating characteristic curve; CI, confidence interval; HDL-C, high-density lipoprotein cholesterol; TG, triglycerides; HbA1c, hemoglobin A1c; GPT, glutamic pyruvic transaminase.

**Table 3  Linear discriminant functions constructed from five major variables for discriminating GADA+ from T2DM patients.**

| Variables | Standardized canonical discriminant function coefficient[a] | Discriminant loading (Structure matrix) | Classification function coefficients (Fisher's linear discriminant functions) | |
|---|---|---|---|---|
| | | | GADA+ | T2DM |
| Age at onset <30 years | 0.46 | 0.37 | 5.556 | 2.208 |
| Age at onset: 30–50 years | 0.36 | 0.11 | 4.274 | 2.576 |
| BMI <23 kg/m² | 0.63 | 0.63 | 6.030 | 2.371 |
| BMI 23–25 kg/m² | 0.22 | −0.11 | 3.756 | 2.421 |
| TG ≥ 98 mg/dL | −0.34 | −0.47 | 2.525 | 4.298 |
| HbA1c ≥ 8.6% | 0.32 | 0.30 | 2.442 | 0.911 |
| HDL-C ≥ 46 mg/dL | 0.36 | 0.43 | 4.658 | 2.808 |
| (Constant) | | | −7.917 | −3.764 |

Notes.

[a]Standardized canonical discriminant function: Wilks' lambda = 0.475, $p < 0.001$; eigenvalue 1.106; canonical correlation = 0.725.

Abbreviations: GADA+, Glutamic acid decarboxylase antibody-positive group; T2DM, type 2 diabetes mellitus group; BMI, body mass index; HDL-C, high-density lipoprotein cholesterol; TG, triglycerides; HbA1c, hemoglobin A1c.

**Table 4  Accuracy of the linear discriminant functions for detecting GADA+ patients.**

| | Predicted group membership | | Total |
|---|---|---|---|
| | GADA+ | T2DM | |
| GADA+ | 128 (84.2%) | 24 (15.8%) | 152 (100%) |
| T2DM | 42 (11.7%) | 316 (88.3%) | 358 (100%) |

Notes.

a. 87.1% of the original grouped cases were correctly classified.

Abbreviations: GADA+, Glutamic acid decarboxylase antibody-positive group; T2DM, type 2 diabetes mellitus group.

age at onset <30 years, and HbA1c ≥8.6%. The overall accuracy of the linear discriminant functions in discriminating GADA+ diabetes was 87.1% (Table 4). The sensitivity was 84.2% and the specificity reached 88.3%.

## DISCUSSION

This study demonstrated that measuring serum TG and HDL-C levels at the onset of disease can be used to help discriminate GADA+ autoimmune diabetes from T2DM in adults. The findings were consistent with those of previous studies that reported lower TG and higher HDL-C levels in patients with autoimmune diabetes (*ADOPT Study Group, 2004*; *Roh et al., 2013*). To the best of our knowledge, the present study is the first to focus on TG and HDL-C combined with BMI, age at onset, and HbA1c to develop a practical discrimination tool to discriminate adult-onset GADA+ autoimmune diabetes from T2DM in ethnic Chinese patients. The LADA clinical risk score proposed by *Fourlanos et al. (2006)* showed a sensitivity of 90% and specificity of 71% for detecting LADA diagnosed by GADA. A Swedish study used fasting C-peptide level <0.7 nmol/L to identify autoimmune diabetes; this showed a sensitivity of 89% and specificity of 66% (*Thunander et al., 2012*). In comparison with these previous studies, the tool proposed in the present study focused on discriminating GADA+ rather than LADA from T2DM. In other words, the target population of GADA+ patients in our study included patients with LADA and classic T1DM. The latter are usually leaner and younger at the onset of diabetes.

Similar to those reported previously, age at diabetes onset and BMI were still major discriminators of autoimmune diabetes. In contrast to the LADA clinical risk score of Fourlanos et al., which uses BMI <25 kg/m$^2$ as a cutoff value and does not include patients younger than 30 years (*Fourlanos et al., 2006*), the present study classified both age and BMI into three levels to make the scoring more precise and showed that BMI <23 kg/m$^2$ and age at onset <30 would be better discriminating features of GADA+ autoimmune diabetes in Chinese patients than BMI 23–25 kg/m$^2$ and age 30–50 years. The possible cause of the lower cutoff of BMI in our study may be due to the different characteristics of study subjects. First, our study subjects included patients with LADA and classic T1DM. Many studies have reported that LADA patients could often be overweight or obese and that being overweight or obese is associated with an increased risk of LADA (*Hjort et al., 2018*; *Luk et al., 2019*). Second, we enrolled subjects with age at onset ≥ 20 years. This was different from the studies that enrolled patients with LADA; the latter are often defined as age at onset >30 years (*Buzzetti, Zampetti & Maddaloni, 2017*). Third, the BMI values in our study were obtained from the initial weight data of newly diagnosed diabetic patients who just received short-term treatment and did not regain their weight fully. Finally, the risk of diabetes starts to increase at a BMI of approximately 23 kg/m$^2$ in Chinese, which is lower than the WHO BMI cutoff value used to define an increase in the incidence among Europeans (*World Health Organization, Regional Office for the Western Pacific (WPRO), International Association for the Study of Obesity, International Obesity Task Force, 2000*; *Ko et al., 1999*). Nevertheless, such a low cutoff may harbor the risk of not being able to identify some patients with LADA, particularly in non-Asians and in areas of high prevalence.

Autoimmune diabetes differs from T2DM in several respects. Its pathogenesis is primarily based on insulin deficiency rather than insulin resistance. The present study showed that patients with GADA+ autoimmune diabetes exhibited higher serum HDL-C and lower TG levels, which can be used to discriminate the condition from

T2DM. Although hypertriglyceridemia may occur in undertreated individuals with autoimmune diabetes due to hyperglycemia, intensive glucose control normalizes the lipid abnormalities (*Subramanian & Chait, 2012*; *Guy et al., 2009*). To reduce any interference from hyperglycemia, the lipid data in the current study were collected about 2–12 weeks after commencing treatment for acute hyperglycemia.

Patients with autoimmune diabetes have worse glycemic control than those with T2DM, possibly due to the limited production of endogenous insulin (*Carlsson, 2019*). The present study showed that patients with GADA+ autoimmune diabetes had higher baseline HbA1c values at disease onset than those with T2DM. Paradoxically, the GADA+ patients had healthier lipid levels despite the higher baseline HbA1c values. Hence, baseline HbA1c was another significant discriminator in our linear discriminant functions, in combination with age, BMI, TG, and HDL-C.

This study had several strengths. All the patients were newly diagnosed diabetes, thus their blood lipid and HbA1c levels were less affected by lipid-lowering and anti-diabetic agents. We also excluded patients who had a habit of drinking alcohol. Using the parameters identified in this study within 3 months of the diagnosis of diabetes may help physicians to decide earlier whether to seek a GADA test for a patient.

The study also had several limitations. First, GADA was not measured for the reference group, so it cannot be ruled out that some of the patients in this group had LADA; however, the possibility of misclassification was reduced by the confirmation from two diabetologists that all of the patients in this group had T2DM. Second, this hospital-based study would inevitably exhibit selection bias. The patients who did receive GADA testing were not representatives for the overall diabetes population. Third, we did not exclude smokers from study participation. Although there was no significant difference in the distributions of smokers between the two groups, smoking might reduce HDL levels and hence increase the probability of false negative results (*Jain & Ducatman, 2018*). Fourth, the incidence and clinical features of autoimmune diabetes may vary between regions and ethnicities (*Buzzetti, Zampetti & Maddaloni, 2017*; *Pieralice & Pozzilli, 2018*). In this study, all the research subjects were Chinese; it remains to be verified whether the findings can be applied to other ethnic groups. Fifth, although 87.1% of original grouped cases could be correctly classified by the linear discriminant functions developed in this study, these functions were obtained from limited samples; they should be tested in an independent population to verify the validity of the proposed method. Finally, although patients with LADA than those with T2DM showed a higher prevalence of family history or personal history of related autoimmune diseases (*Fourlanos et al., 2006*; *Zampetti et al., 2012*), we did not have complete data to include them into our model. From another perspective, the tool requires less information than that needed for calculating Fourlanos et al.'s LADA clinical risk score, as well as a lesser requirement for judgment about acute symptoms at onset and a history of related autoimmune diseases; therefore, our tool may be more efficient in clinical practice.

However, to date, due to the broad heterogeneity of LADA, islet-cell antibodies measurement remains essential to decrease the number of misdiagnosis of diabetes.

Moreover, to detect GADA+ diabetic patients is not the final goal. The clinical problem concerned should be whether insulin-dependency can be detected early by all means.

## CONCLUSIONS

An early diagnosis of adult-onset autoimmune diabetes would avoid patients receiving unnecessary oral medication, as well as helping to predict the disease course and reducing acute events such as DKA. The present study established a convenient diagnostic tool based on data obtained from existing routine tests for diabetes care, including BMI, TG, HDL-C, age at onset, and HbA1c, which may assist with the early identification of adult-onset GADA+ autoimmune diabetes, avoiding extra medical costs and reducing the burden to patients.

### Funding
The authors received no funding for this work.

### Competing Interests
The authors declare there are no competing interests.

### Author Contributions
- Hon-Ke Sia conceived and designed the experiments, performed the experiments, analyzed the data, prepared figures and/or tables, authored or reviewed drafts of the paper, and approved the final draft.
- Shih-Te Tu and Pei-Yung Liao performed the experiments, authored or reviewed drafts of the paper, and approved the final draft.
- Kuan-Han Lin analyzed the data, authored or reviewed drafts of the paper, and approved the final draft.
- Chew-Teng Kor analyzed the data, prepared figures and/or tables, authored or reviewed drafts of the paper, and approved the final draft.
- Ling-Ling Yeh conceived and designed the experiments, analyzed the data, prepared figures and/or tables, authored or reviewed drafts of the paper, and approved the final draft.

### Human Ethics
The following information was supplied relating to ethical approvals (i.e., approving body and any reference numbers):

The study was approved by the Institutional Review Board of Changhua Christian Hospital (CCH IRB No: 190702).

### Data Availability
The dataset is available as Supplemental File.

## Supplemental Information

Supplemental information for this article can be found online at http://dx.doi.org/10.7717/peerj.8610#supplemental-information.

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
