# Peer review of "A convenient diagnostic tool for discriminating adult-onset glutamic acid decarboxylase antibody-positive autoimmune diabetes from type 2 diabetes: a retrospective study"

_PeerJ, doi:10.7717/peerj.8610_

## Round 0.1 · original submission · Major Revisions

The authors should address the concerns raised by the reviewers. Major flaws are in relation to the experimental design (selection of the study population) and validity of the findings.

Reviewer 1 ·

Basic reporting

the aim of the present paper was to develop a convenient tool to discriminate adult-onset GADA-positive autoimmune diabetes from type 2 diabetes in patients with newly diagnosed diabetes. This retrospective cross-sectional study was conducted into n=152 GADA+, and n=358 T2DM.
The Authors identified five major factors discriminant functions: BMI, age at onset, TG, HDL-C, and HbA1c. BMI <23 kg/m2 was the most important factor, followed by TG <98 mg/dL, HDL-C ≥46 mg/dL, age at onset <30 years, and HbA1c ≥ 8.6%.
There are several points that should be clarified.

Major points.
1. The introduction section could be more exhaustive including also the mention, in addition to GADA, of the other islet autoantibodies used to identify LADA patients (IA-2, ZnT8...) and also to the possibility that LADA patients are overweight and obese.( please refer and mention a recent review, i.e. Buzzetti R e at Nat End. Rev, 2017).

2. The Authors reported that they have transformed all the continuous variables with statistical significance. They should specify the reasons why. Only the investigated variables not normally distributed should be transformed. Did they use a test to verify the normality of their data? They should report which kind of test they used.

3. The Authors reported, as an important limitation of the study, that GADA were not measured in the type 2 diabetes patients, but in materials and methods they affirmed that patients with GADA titer ≥1.0 U/mL were excluded from type 2 diabetes. Could they clarify this matter? if GADA were not measured in type 2 diabetes this could be a limitation of the study.

4. Could the Authors specify more clearly the which were the exclusion criteria for the selection of the patients in their study? They reported that patients with an excessive alcohol consumption were excluded. How they evaluated the alcohol consumption?

5. The Authors should show the ROC curve figures and not only the tables relative to the ROC curves..

6. They reported that BMI <23 kg/m2 was the most important factor to discriminate GADA+ patients from type 2 diabetes. This cut off of BMI is very low. Considering such a low cut off there is the risk of not be able to identify correctly patients with LADA. Many papers reported that LADA patients could often be overweight or obese. Recently Hjoirt et al found the importance of overweight/obesity and the risk of LADA (Diabetologia. 2018; 61(6): 1333–1343.

7. Furthermore and more relevant, this low BMI and the young age of onset suggest a T1DM diagnosis! How the Authors can exclude that a substantial group of patients, particularly those in 20-29 age range could be classical T1DM? How did they differentiate LADA from T1DM? Which were the diagnostic parameters to identify a patient as having LADA instead of T1DM?

8. Could the Authors specify using their diagnostic tool (BMI, age at onset, TG, HDL-C, and HbA1c) the number of patients with high GADA titer and low GADA titer identified.

9. Some patients used statins and this treatment could modify the lipid profile, therefore the two groups should be adjusted for use of statins.

10. The Authors conclude that routine tests in diabetes care should be used to establish a convenient, low-cost tool that may assist in the early identification of adult-onset GAD+ autoimmune diabetes in clinical practice. However, to date, due to the broad heterogeneity of LADA, islet-cell antibodies measurement remains essential in order to decrease the number of misdiagnosis of diabetes. The Authors should underline this point in the discussion section.
11. The Authors did not consider two relevant parameters that were took into account by Fourlanos et al in their tool: the presence of other autoimmune endocrinopathies in the LADA patient and a family history for autoimmune diseases. The Authors should mention why, also considering the high prevalence of autoimmune diseases in LADA patients, please refer to the following study Zampetti S et al JCEM, 2012 .

Minor points
1. Line 154 replace “GAGA” with “GADA”.

2. They should replace “sex” with “gender” in the text and table.

Experimental design

see above

Validity of the findings

see above

Additional comments

see above

·

Basic reporting

The topic of the manuscript is to test the hypothesis that GADA-status can be precisely predicted by measuring standard clinical data such as BMI, age at onset of diabetes and standard biochemistry such as HbA1c and HDL.

It is acknowledged that GADA-measurement is not accessible in all settings. However, predicting GADA positive people is not a purpose of it-self. The clinical question to ask is whether the determined variables can predict insulin-dependency as good as (or even better than) than can GADA-measurement. The paper does not provide this answer nor discusses the topic.

Experimental design

A major issue is that the patients who did receive a GAD-measure are not necessary representative for the overall diabetes population unless all newly referred patients had the measure performed in the full study period. One may anticipate that GAD was taken primarily in the patients with higher suspicion of having and autoimmune diabetes.

Validity of the findings

The findings should be tested in an independent population to allow us to test the validity of the finding

Additional comments

The discussion section is much too long and not focused on the topic of the manuscript. For instances line 186-196 discussing mechanisms of insulin resistance is irrelevant
The age cut-of is somewhat arbitrary. GAD-positive patients in the present study will include young lean typical T1D patients whom are diagnosed in the absence of DKA.
L 143: “This study included a total of 510 patients, 152 in the GADA+ group and 358 in the T2DM 144 group”. This information is already given in the method section
145: GAGA+ group: correct typo
Overall I find the paper of modest interest. If the journal accepts short communication the results will be suitable for such a publication

---

## Round 0.2 · accepted · Accept

The authors have satisfactorily addressed all the issues raised by the reviewers.

Reviewer 1 ·

Basic reporting

the authors have answered all the questions raised

Experimental design

it is fine

Validity of the findings

it is fine

Additional comments

the authors have answered all the questions raised

·

Basic reporting

The comments and concerns raised has been met satisfactorily

Experimental design

The comments and concerns raised has been met satisfactorily

Validity of the findings

I still believe that the manuscript would improve significantly if the suggested algorithm for predicting GAD-positivity was tested in an independent population. I know that is a lot of work, but as it stands the results are not readily implementable.

Additional comments

The comments and concerns raised has been met satisfactorily, although rather defensive.